# Same Wavelength Noise Filtering via Quantum Orbital Angular Momentum Emission

**DOI:** 10.3390/s23167118

**Published:** 2023-08-11

**Authors:** Fan Jia, Zijing Zhang, Longzhu Cen, Yanhui Sun, Yuan Zhao

**Affiliations:** School of Physics, Harbin Institute of Technology, Harbin 150001, China; 19b911037@stu.hit.edu.cn (F.J.); 1121120119@hit.edu.cn (L.C.); syhuiii@163.com (Y.S.)

**Keywords:** laser active detection, quantum orbital angular momentum, high signal-to-noise ratio

## Abstract

In laser active detection, detection performance is affected by optical noise, laser interference, and environmental background interference. Conventional methods to filter optical noise take advantage of the differences between signal and noise in wavelength and polarization. Due to the limitations of traditional methods in the physical dimension, noise cannot be completely filtered out. In this manuscript, a new method of noise filtering based on the spatial distribution difference between the quantum orbital angular momentum beam and the background noise is proposed. The use of beams containing quantum orbital angular momentum can make the signal light have a new physical dimension and enrich the information of emitted light. We conduct a complete theoretical analysis and provide a proof-of-principle experiment. The experimental results are in good agreement with the theoretical analysis results, and there is a signal-to-noise ratio improvement of more than five times in laser active detection. Our method meets the urgent needs of laser active detection and can be applied in the field of high-quality target detection.

## 1. Introduction

Laser active detection is a technology that uses laser as a signal carrier to detect information such as intensity, distance, azimuth, altitude, and velocity of a target. Laser active detection has the advantages of high resolution, strong anti-interference ability, and wide application range [1,2,3]. Laser active detection has extensive applications in imaging [4,5,6,7], communication [8], and metrology [9]. At present, the main problem with traditional laser active detection is that the signal-to-noise ratio is affected by noise interference from the same wavelength backscattered by the transmission medium (atmosphere) [10,11], background noise such as sunlight, resulting in the signal-to-noise ratio of laser active detection not meeting the requirements. Currently, research on improving the signal-to-noise ratio is generally focused on processing at the detection receiver. The existing interference filter technology and post-processing noise filtering methods cannot break the limit of classical detection of the signal-to-noise ratio [12,13,14]. The special problem is that there is no filtering method for noise at the same wavelength as the emitted laser. This same wavelength noise is mainly caused by the backscattering of sunlight and the transmission medium.

In the research field of quantum orbital angular momentum modulation generation technology, researchers have carried out research on the method of quantum orbital angular momentum generation and tomography using a Q plate that is electrically Q-switched [15,16]; research on the direct observation of the angular momentum of the quantum orbit of the terahertz beam using the terahertz spiral phase plate [17]; and research on the orbital angular momentum mode multiplexer based on a bilayer concentric micro-ring resonator [18]. In the field of applied research of quantum orbital angular momentum, researchers have carried out rotational Doppler velocity measurement based on orbital angular momentum [19]; research on the weak detection scheme of micro-rotation based on photon angular momentum [20]; and research on the fiber structure for converting a Gaussian beam into the high-order optical orbital angular momentum mode [21]. In addition, research on structured light has also been proposed in recent years [22,23,24,25], and a series of achievements have been made [26,27].

In this manuscript, a detection signal-to-noise ratio enhancement method based on quantum orbital angular momentum modulation emission is proposed. The dimension of emitted light is increased by controlling the emission of quantum orbital angular momentum, so as to effectively distinguish the transmitted signal light and background light noise. This difference is used to filter out the background noise and improve the detection signal-to-noise ratio. To solve the problem of the topological charge attenuation of the quantum orbital angular momentum, this method does not measure the topological charge of the quantum orbital angular momentum that detects the echo signal during detection. The spatial light intensity distribution of the detection signal is utilized to maintain a circular feature distribution of middle dark and surrounding light, while the spatial light intensity distribution of the same wavelength noise is a Gaussian distribution of middle light and surrounding light.

This article designs a noise filtering device based on the spatial distribution difference between signal strength and noise strength. Our detection system uses a spiral phase plate to adjust and generate a beam containing orbital angular momentum as the emission light source. After the light field acts on the target reflection, the noise is filtered through our designed spatial noise filtering device to improve the signal-to-noise ratio of laser active detection. It plays an important role in improving the signal-to-noise ratio of laser active detection and the performance of laser active detection systems.

## 2. System Description and Method

The laser active detection system based on the quantum orbital angular momentum control emission is shown in Figure 1. Laser emission is from the laser, which is modulated into a beam containing quantum orbital angular momentum by the quantum orbital angular momentum regulator after beam expansion. In Figure 1, ③ is the quantum orbital angular momentum regulator, which is made of a spiral phase plate. The light emitted by the laser is modulated by the spiral phase plate, and then the spiral phase is added to obtain the laser beam containing quantum orbital angular momentum. After being transmitted by the transmission system, the target is detected, and the target information is returned to the detection system for reception. ⑥ in the detection system is a quantum orbital angular momentum noise filter, which uses the spatial distribution difference of noise and signal to achieve same wavelength noise filtering. The detector selects a highly sensitive detector to process the filtered signal and complete the detection.

The Laguerre–Gaussian beam is the most typical beam carrying quantum orbital angular momentum in the laboratory and in applications. The expression of the quantum orbital angular momentum modulation signal is obtained by solving the Helmholtz equation. In free space, the electromagnetic field is obtained by solving the scalar Helmholtz equation [27,28]:(1)$$(\nabla^{2}+k^{2})E=0$$
where k is the wave number, k=2π/λ, and E is the electric field.

Using a cylindrical coordinate system (r,θ,z), under paraxial conditions, the electromagnetic field has the form of the following solution:(2)E(r,θ,z,t)=exp[i(kz−ωt)]u(r,θ,z)
where ω is the frequency, t is the time, u is the amplitude function of the electric field.

$l_{p}^{m}\left(x\right)$ refers to normalized Laguerre polynomials, satisfying the following relationship:(3)$$x\frac{d^{2}L_{p}^{1}}{dx^{2}}-(1+1-x)\frac{dL_{p}^{1}}{dx}+pL_{p}^{1}=0$$

Assuming that the Laguerre–Gaussian beam propagates along the *z* axis and concentrates at z, the expression for its complex amplitude is:(4)$$\begin{array}{c} E_{p}^{l}(r,\theta ,z)=\frac{D_{\varphi }}{\omega }{\left(\frac{\sqrt{2}r}{\omega }\right)}^{\left|l\right|}L_{p}^{\left|l\right|}\left(\frac{2r^{2}}{\omega^{2}}\right)\exp\left(-\frac{r^{2}}{\omega^{2}}\right)\exp\left[\frac{ikr^{2}z}{2\left(z^{2}+z_{R}^{2}\right)}\right]\\ \exp\left[-i(2p+\left|l\right|+1)\varphi (z)\right]\exp(il\theta ) \end{array}$$
where $z_{R}=k\omega_{0}^{2}/2$ is the Rayleigh length, $\omega_{0}$ is the waist radius, $\omega =\omega_{0}\sqrt{1+z^{2}/z_{R}^{2}}$ is the beam radius at z, $R=z/\sin^{2}\varphi$ is the curvature radius of the beam at z, $D_{lp}=\sqrt{2p!/\left[\pi \left(p+\left|l\right|\right)!\right]}$ is the normalization coefficient, $L_{p}^{\left|l\right|}$ is the associated Laguerre polynomial. l and p are characteristic quantum numbers representing the mode. p represents the number of rings, and the topological charge l represents the size of the light intensity ring that affects the angular momentum of the photon orbit due to the phase change.

The expression of light intensity distribution of quantum orbital angular momentum is deduced, and Laguerre polynomials $L_{p}^{l}\left(x\right)$ can be expressed as:(5)$$L_{p}^{l}\left(x\right)=\sum_{k=0}^{p}{\left(-1\right)}^{k}\frac{\left(l+p\right)!}{k!\left(p-k\right)!\left(l+k\right)!}x^{k}$$

Bring Laguerre polynomials $L_{p}^{l}\left(x\right)$ into Equation (4). The amplitude expression can be obtained:(6)$$\begin{array}{cc} E_{p}^{l}\left(r,\theta ,z\right)\;= & \frac{1}{\omega }\sqrt{\frac{2p!}{\pi \left(p+\left|l\right|\right)!}}{\left(\frac{\sqrt{2}r}{\omega }\right)}^{\left|l\right|}L_{p}^{\left|l\right|}\left(\frac{2r^{2}}{\omega^{2}}\right)\exp\left(-\frac{r^{2}}{\omega^{2}}\right)\exp\left[\frac{ikr^{2}z}{2\left(z^{2}+z_{R}^{2}\right)}\right]\\ & \exp\left[-i\left(2p+\left|l\right|+1\right)\phi \right]\exp\left(il\theta \right) \end{array}$$

The expression of the light intensity of quantum orbital angular momentum can be obtained through calculation:(7)$$I_{l}\left(r,\theta ,z\right)=\frac{2}{\pi \left|l\right|!}\frac{1}{\omega^{2}}{\left(\frac{\sqrt{2}r}{\omega }\right)}^{2\left|l\right|}\exp\left(-\frac{2r^{2}}{\omega^{2}}\right)$$

From Equation (7) above, it can be seen that the spatial distribution of the signal light field emitted based on the control of quantum orbital angular momentum is annular spatial distribution (hollow annular distribution). The spatial distribution of the background noise light field can be considered as a uniform distribution (medium real distribution). Using the difference in the distribution of light fields between the two, we designed a noise filtering structure as shown in Figure 2. The noise filtering structure consists of a circular transparent diaphragm, with the white part being the transparent part, and the detection signal distributed in a hollow ring passes through. The brown part is the opaque part. The medium real distribution noise is blocked and absorbed, achieving noise filtering. The size of the center radius R of the transparent ring is a fixed value, and the width r of the transparent ring is designed to meet the noise filtering requirements.

The quantum orbital angular momentum is used to control the spatial distribution difference between the light field and the noise, that is, the quantum orbital angular momentum control light field has a circular distribution (hollow circular distribution), and the noise has a uniform distribution (solid distribution), so we use a ring noise filtering aperture to filter out most of the noise. The noise has a uniform distribution (solid distribution), which contains the noise with the same wavelength as the detection signal caused by the backscattering of sunlight and the transmission medium. When implementing noise filtering through our designed spatial noise filtering device, due to the uniform distribution of noise, noise of the same wavelength caused by backscattering of sunlight and the transmission medium is also included in the intensity distribution. So, the method proposed in this article simultaneously and effectively achieves the filtering of same wavelength noise during the noise filtering process.

## 3. Simulations

The process of noise filtering at the same wavelength of the transmitted signal controlled by the topological charge *l* = 1 quantum orbital angular momentum affected by noise is simulated.

The calculation method for the signal-to-noise ratio is to set the incident signal power as $P_{S}$ and the noise power as $P_{N}$. By integrating the light intensity to obtain the power, the signal-to-noise ratio of the system can be expressed as:(8)$$SNR=10\mathrm{lg}(P_{S}/P_{N})$$

Simulation software was used for noise filtering simulation, and the simulation input parameters are the following: laser wavelength 532 nm, waist radius ω_0_ = 3 mm, laser pulse width of 25 ns, and laser emission power of 100 W. The laser emitted by the laser is converted into a quantum orbital angular momentum signal with topological charge *l* = 1 through the spiral phase plate for spatial transmission and is received by the receiving system after being reflected by the target. The simulation results are shown in Figure 3. The pulsed laser is converted into a photonic orbital angular momentum signal with topological charge *l* = 1 through the spiral phase plate for spatial transmission and is received by the receiving system after being reflected by the target. The background light noise is a solid beam, and the signal light is a bright ring beam. According to the above signal-to-noise ratio formula, the signal-to-noise ratio (SNR) of the signal-to-noise beam affected by background light noise is 1.87 (before using the method proposed in this paper for noise filtering).

We set an annular mask plate with a radius of R = 1.25 mm for signal noise separation. The simulation results of the quantum orbital angular momentum modulation detection signal after signal noise separation are shown in Figure 4. The simulation results of the photon orbital angular momentum modulation signal after signal noise separation are shown in the figure. The bright spot of the central background light noise is filtered by the mask plate, and the signal is retained.

After calculating the signal-to-noise ratio formula, the signal-to-noise ratio (SNR) of the same wavelength filtered beam is 13.01. Our method filters out the same wavelength noise and improves the signal-to-noise ratio by 6.9 times.

## 4. Experiment Results

The experimental setup of the imaging system is shown in Figure 5. The laser source of the experimental device of the imaging system is a pulsed laser. The central wavelength of the emitted light is 532 nm, the waist radius of the laser ω_0_ = 3 mm, the mode structure of the laser is TEM00 > 95%, and the root mean square noise is less than 1%. The laser has stable power, good uniformity of the laser field, and uniform speckle after beam expansion. It is a high-performance laser source. In the experiment, we used incandescent lamps as the simulated noise, which is a broad-spectrum noise that contains noise of the same wavelength as the emitted laser detection signal.

The noise filtering process at the same wavelength of the emission signal modulated by the topological charge *l* = 1 quantum orbital angular momentum affected by noise is experimentally studied. Set the target and irradiate the simulated same wavelength noise with an incandescent lamp within the first 6 m of the detection system. The intensity distribution of the noise-scattering light field is shown in Figure 6. The intensity distribution of the background noise light field is Gaussian-like, without circular characteristics, and has a spatial distribution difference from the signal.

Place the target in the field of view of the detection system, about 6 m away from the detection system, and simulate the same wavelength noise by irradiating incandescent light between the target and the detection system. The target echo signal and noise signal are simultaneously collected by detector. In the absence of noise filtering, the light field distribution detected by the detector is shown in Figure 7. The target echo signal and noise signal coexist, and it can still be seen that the target echo signal is a hollow circular light spot, while the noise is a Gaussian-like distribution light spot. The two are superimposed together but still maintain their respective characteristics.

Use our method to read and record the light intensity distribution received by the detector after noise filtering and obtain the light intensity distribution with noise filtering as shown in Figure 8.

According to the calculation of experimental data, the signal-to-noise ratio of the same wavelength noise filtering method for active laser detection based on quantum orbital angular momentum modulation emission is 1.22, while the signal-to-noise ratio of unfiltered noise detection is 0.242.

The experimental results show that the transmitted light can be modulated by the quantum orbital angular momentum, and the signal and noise can be separated according to the difference in the spatial light intensity distribution of the signal and noise. Our method is based on the angular momentum modulation of quantum orbital emission to filter noise of the same wavelength. Under the same conditions, the laser active detection signal-to-noise ratio using the noise filtering method proposed in this paper has been increased by more than five times. See Table 1.

## 5. Discussion

We have demonstrated through experiments a method that can achieve almost perfect noise filtering. Unlike previous methods, orbital angular momentum is chosen as a special degree of freedom to generate spatial distribution differences between signals and noise. By utilizing this difference, signals and noise can be effectively separated. The principle verification experiment has proven the effectiveness and correctness of this method. The main reason for the difference between the experimental results and the simulation results is that the simulation process is ideal, and the orbital angular momentum beam will be distorted with the transmission process in the experimental process, so there is a certain impact on the experimental results.

## 6. Conclusions

In this manuscript, the existing laser active detection method is improved, and the quantum orbital angular momentum method is used to modulate the emission light, so that the signal light has a new physical dimension, and the information carried by the emission light is increased. This difference is used to filter the background noise, and the corresponding noise filtering system is used at the receiver to filter the noise, and the signal and noise are separated according to the spatial light intensity distribution difference between the signal and noise. This realized the filtering of noise at the same wavelength as the emitted laser during the detection process, to achieve the goal of improving the signal-to-noise ratio. In this paper, a laser active detection system based on quantum orbital angular momentum modulation emission is designed. The theoretical research on the modulation emission transmission and reception process of quantum orbital angular momentum in the laser active detection system is carried out, and the simulation and experimental research are completed. The research results indicate that the method proposed in this paper achieves the filtering of noise at the same wavelength in laser active detection, with a signal-to-noise ratio improvement of more than five times, achieving a performance improvement of the laser active detection system.

## Figures and Tables

**Figure 1 sensors-23-07118-f001:**
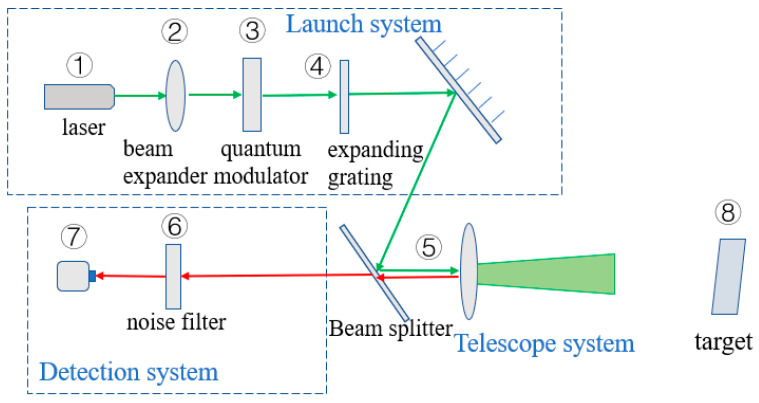
Laser active detection system via quantum orbital angular momentum control emission. (1) Laser. (2) Beam expander. (3) Quantum orbital angular momentum regulator. (4) Beam expanding grating. (5) Emission system. (6) Quantum orbital angular momentum noise filter. (7) Detector. (8) Target.

**Figure 2 sensors-23-07118-f002:**
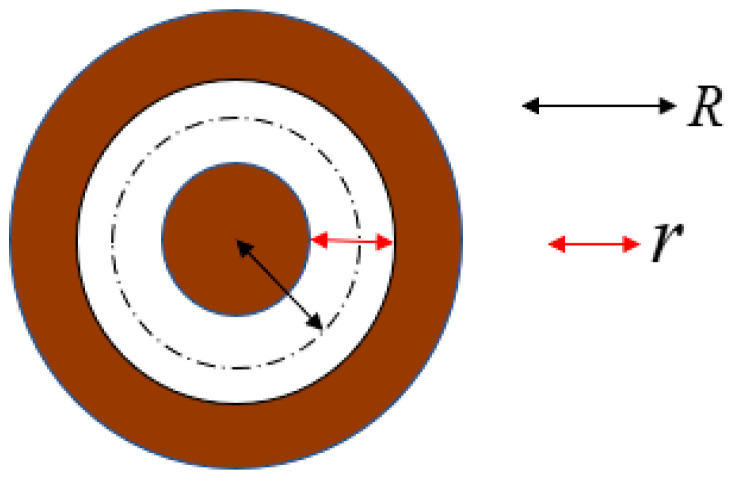
Structure diagram of quantum orbital angular momentum noise filter.

**Figure 3 sensors-23-07118-f003:**
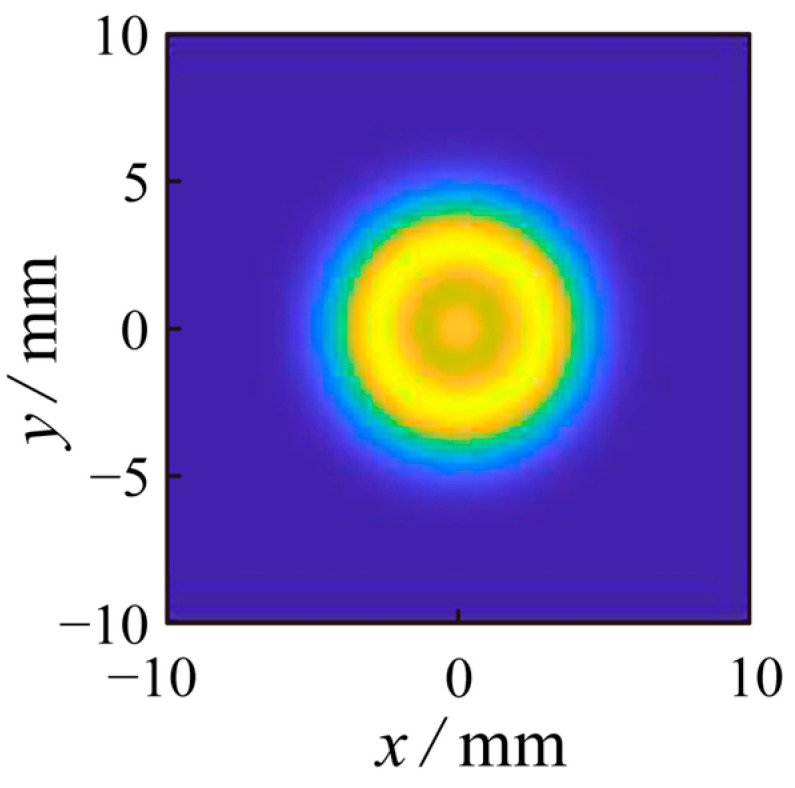
Optical intensity distribution diagram of quantum orbital angular momentum signal with topological charge *l* = 1 affected by noise (before noise filtering by this method).

**Figure 4 sensors-23-07118-f004:**
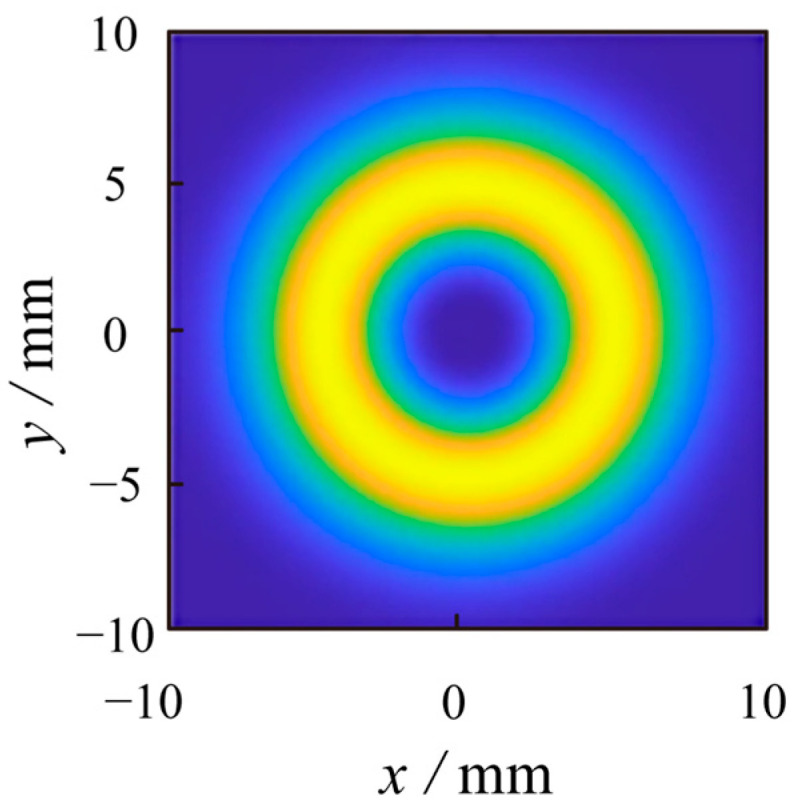
Optical intensity distribution of noise-affected topological charge *l* = 1 quantum orbital angular momentum signal after signal noise separation (after noise filtering by this method).

**Figure 5 sensors-23-07118-f005:**
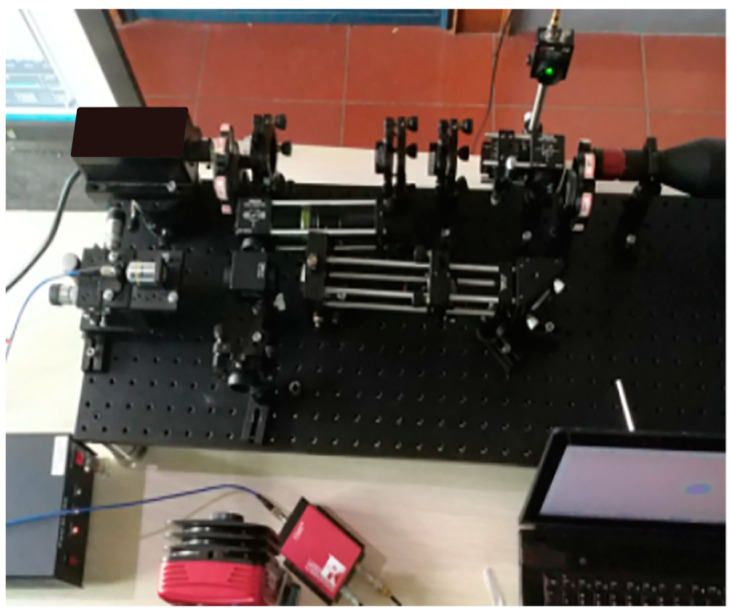
Experimental system for active detection of noise at the same wavelength based on quantum orbital angular momentum control laser.

**Figure 6 sensors-23-07118-f006:**
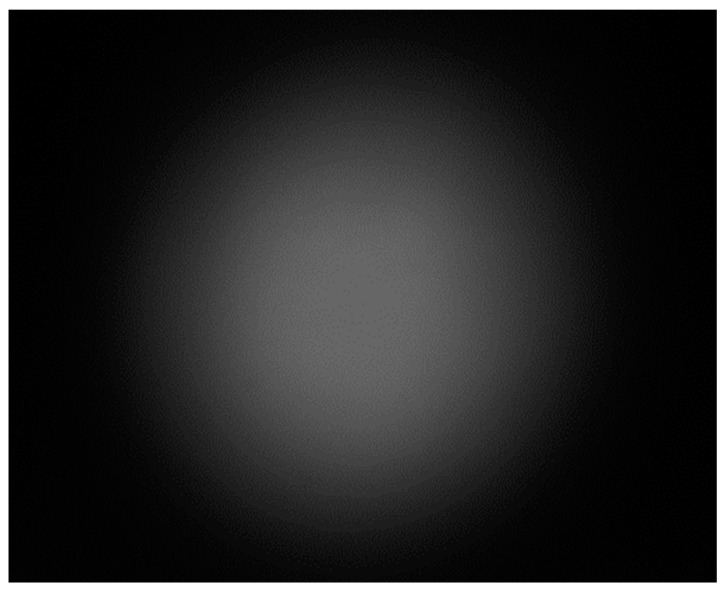
Intensity distribution of same wavelength noise simulated by incandescent lamp irradiation.

**Figure 7 sensors-23-07118-f007:**
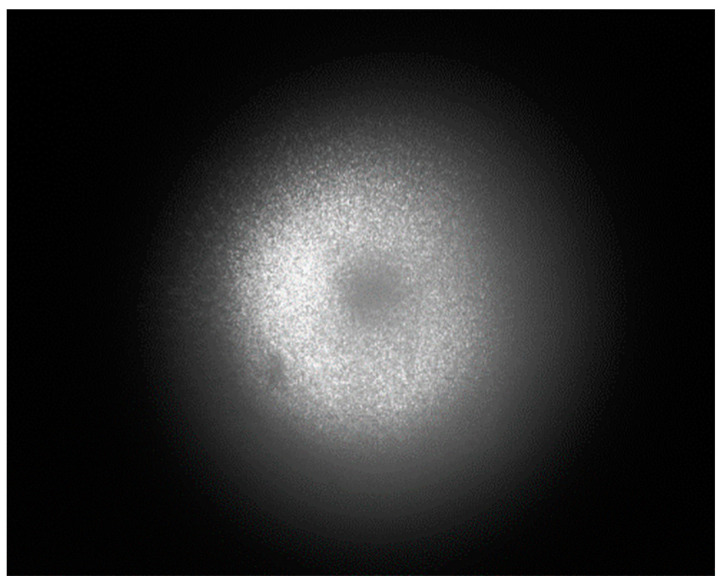
Light field distribution of signal and noise superposition without noise filtering.

**Figure 8 sensors-23-07118-f008:**
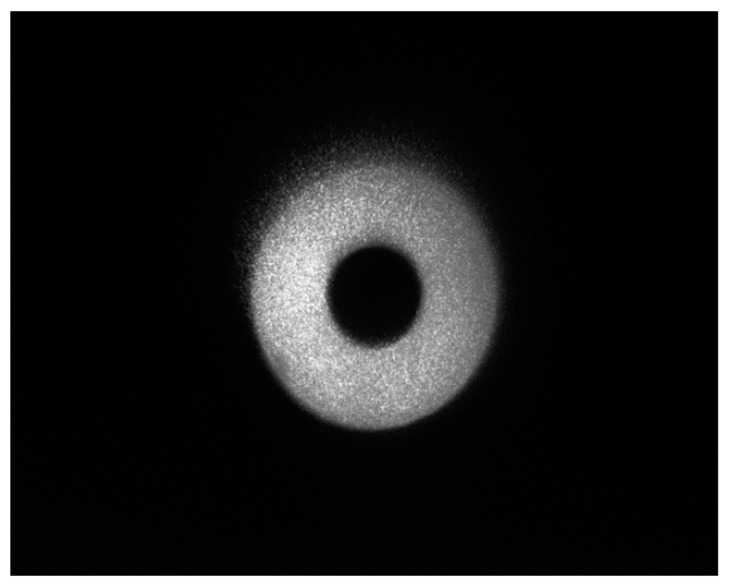
The light field distribution of the signal and noise superposition after noise filtering using the method described in this paper.

**Table 1 sensors-23-07118-t001:** Comparison of simulation and experimental signal-to-noise ratio improvement results.

Result Type (Topological Charge *l* = 1)	Signal-to-Noise Ratio (Before Noise Filtering)	Signal-to-Noise Ratio (After Noise Filtering)	Signal-to-Noise Ratio Improvement	Signal-to-Noise Ratio Improvement Multiple
Simulation	1.87	13.01	8.42 dB	6.9 times
Experiment	0.242	1.22	7.03 dB	5.0 times

## Data Availability

Data available on request due to privacy.

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
