# Peer review of "Same Wavelength Noise Filtering via Quantum Orbital Angular Momentum Emission"

_sensors, 2023, doi:10.3390/s23167118_

Round 1
Reviewer 1 Report
-Abstract is too simple. Should Summarize the results in abstract. Not just mentioned the measurement result better than simulation.
-Please provide table results comparison between simulation and measurement
-Please discuss academically the results instead of reporting the figure in simulation and measurement result section
-most of the referencess more than recent 5 years. Please update
Author Response
On behalf of our coauthors, we would like to thank you again for considering our manuscript and for your careful handling of our submission. As requested, we have addressed each and every single comment and question raised by the referees in a point-by-point fashion below.
We are thankful to the referees for their detailed reports. Indeed, the referees provided comments that enabled us to provide stronger motivations and to include additional calculations in the new version of our manuscript. We are certain that the current manuscript will be well-received by readers interested in the field of laser active detection.
We believe that we have successfully addressed each comment from referees and consequently, we feel that our manuscript is now suitable for publication in Sensors.

Reviewer 2 Report
The manuscript is well organized and the subject is interesting to reduce noise. I think that the novelty of the presented method is sufficient to approach acceptance. Therefore, I recommend it for publication after addressing the following comments:
- The literature review is not updated. The recent reports should be added to the survey.
- Substitute ‘db’ to ‘dB’.
- How the samples of noise are selected for simulation and experiment?
Author Response

(The authors gave the same response as above.)

Reviewer 3 Report
Thank you for your effort.
I have some comments that must be considered in the modified manuscript.
-----------------------------------------------
In this paper, a new method is proposed (theoretically and aided with experiment) for noise filtering based on the spatial distribution difference between the quantum orbital angular momentum beam and the background noise.
Comments:
1) Both (Abstract) and (Conclusion) are qualitative (descriptive). Both need some numerical value(s) of the main finding(s) to be quantitative.
2) Add a paragraph (at the end of "Introduction") for paper organization.
3) In your (Introduction) you explained clearly the related work and after that you mentioned what you did in this paper. Please, clarify what is new (novelty) or what you added in your work that is different than the others.
4) Many equations in the model need references. Ex: Equations 1+2+3+4.
5) Figure 2 needs to describe how this structure can filter noise. Please explain.
6) After Eq. 8, you wrote: Using simulation software.... What software?
7) The values in the paragraph after Fig. 8 need units. It is regular to have SNR in dB.
8) Also, you wrote: Under the same conditions, the laser active detection signal-to-noise ratio using the noise filtering method proposed in this paper has been increased by more than 5 times.
I cannot see this in the graphs. Please clarify how did you get the 5 times?
9) References need some more recent references. You have 1 ref. in 2023 and nothing in 2021 and 2022
Author Response

(The authors gave the same response as above.)

Reviewer 4 Report
This is an interesting paper on filtering the signal of the reflected laser beam from the environment noise light, Authors proposed a method and checked it practically. Potentially the paper is interesting to all Lidar techniques, even if the testing was on a short distance (6 m).
Some imperfections of the style, like in line 74, "The laser emits a laser",
line 235 no need to repeat "signal and noise", just say "them"
I do not understand a statement: beam containing quantum orbital angular momentum by the quantum orbital angular 75 momentum regulator
Photons bring angular momentum, so how it could not contain the angular momentum? Please, reformulate.
Lines 132-137 seem to repeat the same information. Please, check.
Line 182 authors could give the commercial name of the laser.
Some problem with capital letters. Line 102 should read: Laguerre
line 105 (and 221) should read: the angular, line 178: pulsed,
Adjust, please, references (space after commas etc.) What is [J]?
Please make extensive editing, mainly to avoid repetitions of the same/ similar statements.
Author Response

(The authors gave the same response as above.)

Reviewer 5 Report
I can recommend this paper for publication if the authors discuss in the introduction other papers about structured light:
1. Physical Review B 104, 165402 (2021)
2. Nature Photonics 16, 523 (2022)
3. Nature communications 12, 5891 (2021)
Moreover, the structure on figure 2 is similar to proposed in
Phys. Rev. B 107, 155104.
Perhaps the authors will find interest in the ideas proposed in this paper.
)
Author Response

(The authors gave the same response as above.)
